# Exploring Performance Degradation in Virtual Machines Sharing a Cloud Server

**Hamza Ahmed** [1], **Hassan Jamil Syed** [1,2,3,*], **Amin Sadiq** [1], **Ashraf Osman Ibrahim** [2,4], **Manar Alohaly** [5,*] **and Muna Elsadig** [5]

1   FAST School of Computing, National University of Computer and Emerging Sciences, Karachi 75030, Pakistan; k214805@nu.edu.pk (H.A.); k214001@nu.edu.pk (A.S.)
2   Faculty of Computing and Informatics, Universiti Malaysia Sabah, Jalan UMS, Kota Kinabalu 88400, Sabah, Malaysia; ashrafosman@ums.edu.my
3   Cyber Security Research Lab, Faculty of Computing and Informatics, Universiti Malaysia Sabah, Jalan UMS, Kota Kinabalu 88400, Sabah, Malaysia
4   Creative Advanced Machine Intelligence Research Centre, Faculty of Computing and Informatics, Universiti Malaysia Sabah, Jalan UMS, Kota Kinabalu 88400, Sabah, Malaysia
5   Department of Information Systems, College of Computer and Information Sciences, Princess Nourah bint Abdulrahman University, P.O. Box 84428, Riyadh 11671, Saudi Arabia; memohamedahmed@pnu.edu.sa
*   Correspondence: shjamil@ums.edu.my (H.J.S.); mfalohaly@pnu.edu.sa (M.A.)

**Abstract:** Cloud computing has become a leading technology for IT infrastructure, with many companies migrating their services to cloud servers in recent years. As cloud services continue to expand, the issue of cloud monitoring has become increasingly important. One important metric to monitor is CPU steal time, which measures the amount of time a virtual CPU waits for the actual CPU. In this study, we focus on the impact of CPU steal time on virtual machine performance and the potential problems that can arise. We implement our work using an OpenStack-based cloud environment and investigate intrusive and non-intrusive monitoring methods. Our analysis provides insights into the importance of CPU steal time monitoring and its impact on cloud performance.

**Keywords:** virtual machines; cloud computing; CPU steal time; performance degradation; cloud monitoring; OpenStack; resource contention; performance metrics; workload management; scalability; performance optimization

## 1. Introduction

In recent years, many companies have been migrating to cloud services, which rely on shared resources instead of on-premises or individual on-site servers for application management. This shift away from the traditional IT infrastructure is moving organizations beyond the physical boundaries of their premises. As the cloud computing infrastructure expands, monitoring the system for optimal performance becomes increasingly important. By doing so, companies can proactively identify potential issues and mitigate any negative impact on their operations.

Cloud technology meets the organization's needs by offering Infrastructure as a Service (IaaS), Software as a Service (SaaS), and Platform as a Service (PaaS). Monitoring is one of the critical issues in cloud services [1].

The monitoring of a system is an essential need of an organization. Nowadays, this can be performed in many ways, but the critical issue is the delay, which consumes time. In [2], it is shown that monitoring a virtual machine that runs on a single host server and sharing resources with four other VMs on the same host does not mean that they all use an equal 25% of the host server. It is allowed to use more than its proportion. If, at that time, another VM requires memory it has to wait until the hypervisor is scheduling tasks between the VMs; the time that the virtual machine waits for the host server is called steal

time. In [3], a study was conducted on the Amazon EC2 m3 medium model server to determine the delay and its impact on the actual world situation, and the results proved that it is highly affected by the CPU steal time because of the lack of monitoring parameters, which are also impacted by the steal time.

In a cloud compute node, the VMs work as a process. In [1], the work conducted so far concerns the non-intrusive monitoring of how the hypervisor schedules the task. It would greatly facilitate cloud monitoring if we could add information about the monitoring of the CPU steal time. As discussed in [4], several tools exist, but we cannot capture the steal time. Gartner's report [5] predicts that in 2020, the cloud computing share of the market will be USD 266.4 billion, which is an increase of USD 227.8 billion, or 17%, from 2019. Significant growth will be observed in the field of SaaS and will increase to USD 116 billion in 2020 and USD 99.5 billion in 2019. This increase of 16% was the highest. According to the SRVR ETF fact sheet, by the end of 2019 the ETF's top five holdings were Equinix, 15.3%; American Tower, 13.2%; Crown Castle, 14.2%; GDS, 5.7%; and Lamar Advertising, 5.1%. Equinix spends around USD 17 billion per year on data centers, which is more than Google, Microsoft, and Amazon, which spend around USD 10 billion annually. This shows how large companies move their servers to the cloud [6]. This explosive growth in cloud computing will increase the chance of network failure and will cost millions in a few seconds; so, cloud monitoring is much more critical. A recent study showed that around two or more monitoring tools are used by more than 70% of companies to monitor their cloud data [7].

Several problem factors can occur in cloud monitoring. There are several causes of the problem, and to form the problem into the shape of a statement, we must make it more generalized. Firstly, we examined the event's issues and obstacles and found its causes. The monitoring system used currently involves overheads while monitoring the VM. Cloud monitoring systems use agents, also called intrusive cloud monitoring solutions [1,8,9]. An invasive monitoring solution inserts agents; this causes the VM to carry a greater burden due to the agents' involvement. Agent-based monitoring, which utilizes the VM's maximum power, can cause problems when the owner does not wish for its insertion, as it takes all the power because of load and delay factors. Some operations are very severe, and the owner does not want a high level of delay. If this is the case, then this monitoring system for the VM completely fails. Cloud monitoring is always a problem in terms of load, security concerns, and increasing overheads. These factors will improve in time, causing the hypervisor to schedule more tasks between VMs. This will cause an increase in the steal time factor, which we have to monitor. In this research, we aimed to perform cloud monitoring and to monitor the critical elements of the steal time information. The following set of objectives needs to be fulfilled while conducting this research.

- To review the current monitoring solutions and how they can be mapped in the monitoring of steal time.
- To propose a solution by which we can measure steal time information.
- To implement the solution in the cloud to compare it with the existing solutions.

The rest of the article is structured as follows. Section 2 reviews the related works. The architecture of the proposed solution is presented in Section 3. The suggested architecture's implementation phases are elaborated on in Section 4. Section 5 provides a discussion of the results. Finally, we conclude the study in Section 6.

## 2. Related Work

We discuss the research and investigation of problems in cloud monitoring technology and data networks that allow them to fulfill their needs and achieve their goals. There are several challenges involved in keeping control of cloud network resources. A literature review on tracking the performance of cloud networks and their help showed that today, despite the particular considerations and the increased expenses related to guaranteeing the performance of information systems, there are various issues in this field of research. The work in [3] contains a prediction of the performance between high-level and low-level

CPUs using mathematical relationships. The goal is to compare the influence of application time and CPU steal time. The study in [1] discusses the issue related to measuring execution time and reducing it to a minimum. It also discusses the execution time in the non-virtual environment compared to that in the virtual environment. Monitoring is further categorized into two significant types: (i) intrusive monitoring and (ii) non-intrusive monitoring. In [10], on intrusive monitoring, the agents monitor environmental metrics such as performance and accuracy, which have several drawbacks, like latency and overhead. In recent years, some used an intrusive monitoring approach which increases the overheads on VMs. Non-intrusive monitoring data are directly collected from the OS or the cloud—the Amazon EC2 m3 medium model server is highly affected by CPU steal time; a study has shown that in a natural world environment, when the load increases, the steal time factor has a significant impact on the hypervisor [3,11]. Each VM has its OS. Still, with the rise in the number of virtual CPUs at the compute node, the hypervisor cannot schedule the assignment of all the virtual CPUs to physical ones. The time in which the hypervisor organizes the assignment of a physical CPU to a virtual CPU is called steal time. With the increase in virtual CPUs, steal time cannot be neglected [12].

The study in [13] presents a comparative and systematic study of the techniques for secure data sharing and protection in the cloud. Each method is discussed in terms of its functionality, potential solutions, workflow, achievements, scope, gaps, and future directions. It aims to provide insights into the applicability of these techniques and to identify research gaps for future work in the field. Apache Spark [14] is a popular in-memory data analytic framework for various applications. Existing resource allocation strategies rely on user-specified peak demand, which leads to low cloud utilization in production environments. To address this issue, the article presents iSpark, a utilization-aware resource provisioning approach for iterative workloads on Apache Spark. iSpark dynamically adjusts the number of allocated executors based on real-time resource usage, scaling up or down as needed. It also preemptively handles underutilized executors and preserves cached intermediate data for data consistency. Testbed evaluations demonstrate that iSpark significantly improves resource utilization, cluster utilization, and job completion time compared to Vanilla Spark. The study in [15] explores the issue of latency caused by the hypervisor, which can introduce noise into the measured latency of the virtual environment. To address this problem, the author proposes an algorithm to filter out this noise and improve the accuracy of the computed metrics. The algorithm is designed to be hypervisor-agnostic, making it applicable to various virtual environments, whether deployed locally or in cloud services with different hypervisor technologies. An overview is given of hypervisor technologies and their impact on latency when executing processes in virtual environments. By applying the proposed algorithm, the network quality measurements from virtual environments become more reliable, allowing for a better understanding of unexpected latencies. In [16], a two-stage genetic mechanism for load balancing virtual machine hosts (VMHs) in cloud computing is developed. Different methods are focused on the current limitations of VMHs without considering the future loads, limiting their effectiveness in real environments. The proposed approach integrates two genetic-based techniques. Firstly, performance models of virtual machines (VMs) are generated using gene expression programming (GEP), which allows for the prediction of the loads of VMHs after load balancing. Secondly, the genetic algorithm considers both current and future loads of VMHs to optimize the assignment of VMs to VMHs, facilitating load balancing through VM migration. The performance of the proposed methods is evaluated in a real cloud computing environment, Jnet, where they are implemented as a centralized load-balancing mechanism. The experimental results demonstrate that the proposed method surpasses previous approaches, including heuristics and statistical regression, in terms of performance and effectiveness.

Here, how the monitoring data are collected through centralized or decentralized monitoring architecture is explained. In centralized monitoring, there is a chance of a single point of failure, which is not the best choice. In centralized monitoring, the server is run

on a physical machine on the cloud control node; moreover, the updates of every single node to the centralized server increase bandwidth utilization. In decentralized monitoring architecture, the chance of a single point of failure has been eliminated, and each node shares information with its peer when required; thus, there is no need for a single ample storage requirement. Here, the network utilization is also controlled. Different types of clouds were identified while conducting a literature review [17]. A single organization mainly operates a private cloud, and all the operations performed in it are by the organization itself; the public cloud is open and can be used by the local public; the community cloud is shared by a group of people that have the same shared concerns; the hybrid cloud consists of two or more different clouds (public, private, or community). Many researchers have worked on VM monitoring, but the critical issue still remains: the steal time information. In [3], the author emphasized that the importance of CPU steal time is a severe issue as its interference hits the subscriber in terms of time loss; an experiment was performed on Amazon EC2 m3. CPU steal time has seriously impacted medium model instances and the runtime workload. The study says that there is no way this loss can be monitored (i.e., CPU steal time). Time constraint is a severe issue for VMs; without knowing the importance of the work running a specific VM, the CPU delays it; this issue has to be handled very seriously [18]. Cloud monitoring data are collected from a node. There are two types of data collection: timely driven or event-driven. There are two modes: a server will request or the node will send the data. Generally, the three-cloud monitoring solution includes (i) a push model, (ii) a pull model, and (iii) a hybrid model.

VMs and containers in cloud monitoring solutions are part of the operating systems. As the cloud environment speed is too fast, when creating VMs the time is impossible to measure. In [8], the author presents a new monitoring paradigm where no monitoring agent is involved within the host VM. No extra software is used in monitoring with NFM. The author tested over 1000 different types of Linux for NFM; these worked perfectly well. In [19], the author proposed a hybrid push protocol for monitoring. The name hybrid is suggested for this protocol as it uses event-driven and regular push protocols. When using a hybrid protocol, only one metric in the experiment is chosen as this reduces the overheads.

In [20,21], the authors focused on reducing the amount of monitoring data in storage. In grid computing, the data can still be present after the resource is utilized, whereas VMs are terminated because the job is performed in the clouds. Monitoring solutions where an agent is involved cause an extra load, resulting in a heavyweight design [8]. In [22,23], the author proposed the importance of monitoring in order to make reliability and security the critical factors in the monitoring grids. The proposed strategy for creating a research testbed is OpenStack, through which the lab can implement cloud simulation; in [24], it is clearly shown that decision makers do not consider monitoring a severe issue even though they face problems with high application downtime. The monitoring is even conducted in a raw manner using basic technology like MySQL querying; lastly, the big issue is that the clients find problems instead of application owners. In [25], the survey conducted is to highlight and identify the capabilities of monitoring tools. The author shows the comparison and contrast of different monitoring tools. In [26], the author proposed a solution for the Linux kernel that recognizes the increase in CPU steal time and uses it for the scheduled processes and threads. The experimental results in [27] show that the effect of steal time in the Android OS is typically '0'. In [28], the experimental results show the precise impact of steal time on the application runtime.

Table 1 presents an overview of the monitoring architectures' characteristics, including their centralization/decentralization and scalability and a consideration of monitoring CPU steal time. However, the information regarding the monitoring platform overheads is incomplete or not provided for most of the architectures. DARGOS is a scalable, decentralized architecture that considers monitoring CPU steal time. PCMONS has a centralized monitoring architecture and is scalable. It is not applicable (N/A) for monitoring platform overheads, and no information about monitoring CPU steal time is available. GmonE also has a centralized monitoring architecture and is scalable. It does not provide information

on monitoring platform overheads, but it considers the monitoring of CPU steal time. NFM does not specify its monitoring architecture, but it is decentralized and scalable, and it considers the monitoring of CPU steal time. However, there is no information about monitoring platform overheads. MonSLAR is a scalable, decentralized architecture that does not consider the monitoring of CPU steal time. Ngmon is similar to MonSLAR in terms of characteristics. It is decentralized and scalable, and it considers the monitoring of CPU steal time. MonPaaS does not specify its monitoring architecture, but it is decentralized and scalable, and it considers the monitoring of CPU steal time. CloudProcMon has a centralized monitoring architecture and is scalable. It considers both monitoring platform overheads and the monitoring of CPU steal time. No specific information is provided about the scalability of the monitoring platform overheads or the monitoring of CPU steal time.

**Table 1.** Comparative Table.

| Paper Reference | Monitoring Architecture | | Monitoring Platform Overheads | Scalability | Monitoring CPU Steal Time |
|---|---|---|---|---|---|
| | Centralized | Decentralized | Considered | Considered | Considered |
| DARGOS [29] | N/A | ✓ | ✓ | ✓ | N/A |
| PCMONS [30] | ✓ | N/A | N/A | ✓ | N/A |
| GmonE [31] | ✓ | N/A | ✓ | ✓ | N/A |
| NFM [8] | N/A | ✓ | ✓ | ✓ | N/A |
| MonSLAR [32] | N/A | ✓ | ✓ | N/A | N/A |
| Ngmon [33] | N/A | ✓ | ✓ | ✓ | N/A |
| MonPaaS [34] | N/A | ✓ | ✓ | ✓ | N/A |
| CloudProcMon [1] | ✓ | ✓ | ✓ | ✓ | N/A |
| Proposed Solution | ✓ | ✓ | ✓ | ✓ | ✓ |

After reviewing state-of-the-art monitoring solutions and steal time information in detail, this literature review shows that monitoring is a critical issue in all cloud monitoring areas and that the scheduling of tasks between VMs takes time. This crucial issue of steal time needs to be monitored, and we will investigate this issue using cloud monitoring.

## 3. Architecture of the Proposed Solution

The proposed solution is for a cloud monitoring framework to monitor the critical issue of measuring CPU steal time. This monitoring solution has some distinguishing features, including lightweight data collection monitoring, scalability, and automated performance monitoring. Furthermore, we discuss the different algorithms used to obtain the monitoring data of VMs. Also, a discussion on the proposed algorithm for extracting the information on CPU steal time is presented.

Many researchers have proposed different monitoring solutions. Some monitoring solutions, like CloudProcMon [1], work very efficiently. NCP is another monitoring solution that uses cloud component metrics to generate the results. In the literature review, we review several cloud monitoring solutions; Our proposed solution highlights the shortcomings of the existing solutions. Some features of our monitoring solution which help us to monitor CPU steal time are:

(1)  Scalability

A system needs to be dynamic and scalable horizontally and vertically. It needs to be compatible with any changes and to work in any situation. Horizontally scalable means that increasing the number of nodes in the network will work well with it. In contrast, vertically scalable means that the node can handle multiple resources simultaneously.

(2)  Monitoring Performance Automatically

This automatic monitoring feature of the steal time monitoring framework sets it apart from other monitoring systems. It eliminates manual intervention or human interaction, making the monitoring process more efficient and streamlined.

To achieve this, the framework offers two options for monitoring:

(i) Using the push communication model: The monitoring framework actively pushes real-time steal time data to the relevant monitoring entities or systems with this approach. It ensures that the monitoring information is promptly delivered without relying on manual requests or queries. The push model enables continuous monitoring and immediate access to steal time data, allowing for proactive detection and response to performance issues.

(ii) Using the pull model: Alternatively, the monitoring framework supports the pull model, where the monitoring entities can retrieve steal time data autonomously as needed. With this approach, the monitoring systems can periodically request steal time information from the framework, or they can provide it when demanded. The pull model provides flexibility and adaptability in monitoring operations by enabling self-initiated data retrieval.

The architecture of our proposed solution is depicted in Figure 1; it comprises two main components: the cloud controller node and the compute node. The cloud controller node hosts the monitoring dashboard, a user interface for monitoring and visualizing collected data. On the other hand, the compute node represents the cloud server where the virtual machines (VMs) are deployed. The monitoring dashboard establishes communication with the VMs on the compute node to gather specific data, mainly information related to CPU steal time. The collected monitoring data from the VMs on the compute node are transmitted to the monitoring dashboard located on the cloud controller node. These data are then subjected to further analysis and visualization.

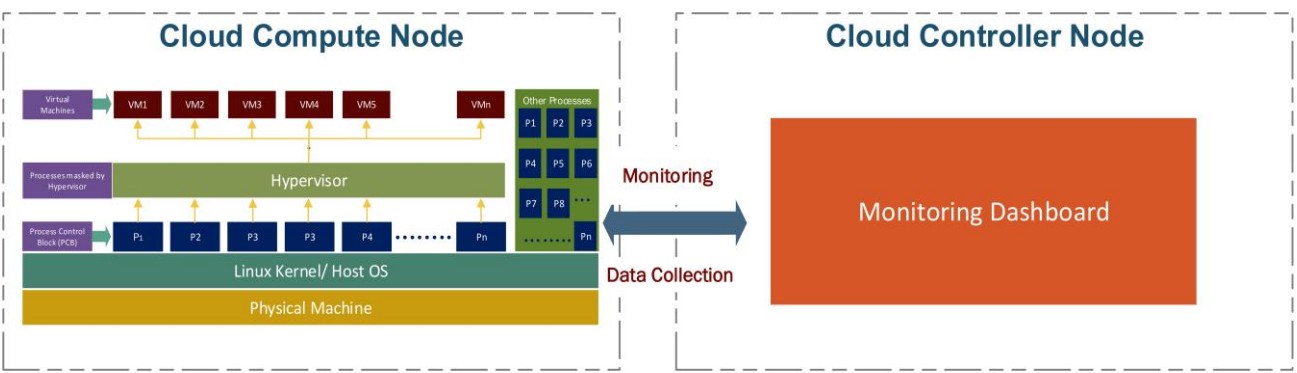

**Figure 1.** Schematic diagram of the proposed solution.

Each VM on the compute node behaves as an individual process and is assigned a unique process ID (PID). To access the current status of and information on the VMs, our monitoring solution utilizes the /procfs folder on the compute node. For the data collection necessary to monitor the CPU steal time of the VMs, our proposed solution incorporates specific algorithms, namely Algorithm 1 and Algorithm 2.

A.    Generalized Overview of the Proposed Solution

We utilized an OpenStack-based cloud monitoring solution consisting of the cloud controller node and the compute node. This monitoring solution effectively captures the stolen time data information from the VMs on the cloud compute node while the controller node hosts the monitoring dashboard. Multiple VMs are launched on the cloud compute node, with each VM behaving independently and possessing a unique process ID. The current status of all the VMs of the host OS is accessible through the /procfs folder, enabling comprehensive monitoring capabilities.

B.    Algorithm(s) used for the Proposed Solution

Here, we discuss the proposed algorithms for our monitoring solution. These algorithms are used by our monitoring solution to collect data. The key issues that our submitted algorithm addresses are as follows.

- The obtaining of the PID of the OS interface.
- The monitoring of the CPU steal time.

The following are the algorithms used to collect the monitoring data for the metrics mentioned above.

- Algorithm 1: To obtain the PID of the VM.
- Algorithm 2: To collect data for the CPU steal time of a VM.

---

**Algorithm 1: OS interface**

---

  1  **Receive** query x from Monitoring.
  2  **Do**
  3  **if (x = PID)** then
  4      **Get** PID from */proc*
  5      *Monitoring* ← PID
  6  **Else**
  7      **if (x = server_ID)** then
  8          **Get** server_ID from */proc/PID/cmdline*
  9          *Monitoring* ← server_ID
10      **end if**
11  **end if**
12  **End**

---

Algorithm 1 represents the OS interface, which plays a crucial role in monitoring. When the monitoring component sends a query (denoted as x) to the OS interface using the push model, the algorithm processes the query and responds accordingly.

The algorithm begins by receiving the query x from the monitoring component. It then proceeds to evaluate the query by following the steps outlined below. If the query x requests the process identifier (PID), the algorithm enters the "if" condition at line 3. In this case, it retrieves the process identifier (PID) from the /proc directory, which contains system information about the running processes. The obtained PID value is then assigned to the monitoring component, enabling it to track and monitor the specific strategy. Alternatively, if the query x pertains to the server identifier (server_ID), the algorithm moves to the "else if" condition at line 7. It retrieves the server_ID associated with the queried process by accessing the /proc/PID/cmdline file. This file contains the command line arguments to launch the process with the specified PID. The algorithm retrieves the server_ID value from this file and assigns it to the monitoring component, allowing it to monitor and track the specific virtual machine (VM) within the cloud environment. Utilizing Algorithm 1, the OS interface bridges the monitoring component and the operating system. It effectively handles different queries, retrieves the relevant information from the appropriate system files, and delivers the necessary data back to the monitoring component, facilitating comprehensive monitoring capabilities.

Algorithm 2 outlines the process of monitoring the CPU steal time for a virtual machine (VM). The CPU steal time refers to the amount of time a virtual CPU waits for the physical CPU. This algorithm collects the CPU steal time statistics and evaluates them against predefined threshold values to determine the status of the VM. The algorithm begins by starting the process (line 1). It then retrieves the process ID (PID) of the VM from the /proc/directory of the host machine (line 2). Using this PID, the algorithm accesses the /proc/PID/stat file to obtain the CPU steal time of the VM (line 3). The algorithm enters a loop (lines 4–13) to compare the CPU steal time against specific threshold values and to assign a corresponding status. The status is set to 'OK' if the CPU steal time is less than 2 (lines 5–6). If the CPU steal time is between 3 and 7, the status is set to 'WARNING' (lines 7–8). If the CPU steal time exceeds 7, the status is 'CRITICAL' (lines 9–10). After determining the status, the algorithm prints a message indicating the CPU steal time of the VM along with the corresponding status (line 12). This provides visibility to the current CPU steal time performance of the VM. The loop continues (line 13) if the CPU steal time exceeds 0, allowing for ongoing monitoring and status evaluation. By employing this algorithm, one can effectively monitor the CPU steal time of a VM and receive status

updates based on predefined thresholds, enabling the proactive identification of potential performance issues.

---

**Algorithm 2: CPU steal time**

---

```
1   Start
2   Get the PID (the process ID of the Virtual Machine) of VM from /proc/ of the host machine.
3   Get CPUstealtime of the VM using /proc/PID/stat
4   Do
5              if (CPUstealtime < 2) then
6                  status = 'OK'
7              Else if (CPUstealtime > 3 AND CPUstealtime < 7) then
8                  status = 'WARNING'
9              Else if (CPUstealtime > 7) then
10                 status = 'CRITICAL'
11             Endif
12                 Print "CPUstealtime of VM is, CPUstealtime, '%'."
13  while CPUStealtime > 0
14  End
```

---

C.    Steal Time Monitoring Code

Steal time monitoring data can be collected using bash scripts. Here, we discuss the code used in our monitoring solution. The code listed below is divided into subparts, and each metric is discussed separately. The following are the code strategies we used in the data collection.

First, we have to figure out the process ID of the VM for which we are calculating the steal time information. Listing 1 shows how the PID is collected from the procfs folder.

**Listing 1**: PID.

$$PID = \$(pgrep\ qemu\ |\ head\ -1\ |\ tail\ -1)$$

When this line of code is executed on the VM, we obtain the 'PID' of this VM and then have to collect the CPU steal time for the VM. Listing 2 shows the code to obtain the CPU steal time information of the VM. This information is stored in the variable called 'steal time'.

**Listing 2**: CPU Steal time.

$$stealtime = \$(top\ -b\ -n\ 2\ |\ head\ -n\ 3\ |\ tail\ -n\ 1\ |\ awk\ '\{print\ \$16\}')$$

Now, we collect the CPU steal time of each VM. In the last part of our code, we have to set some threshold values for the monitoring data display results.

We developed some cases in which the first case shows the result 'OK' if the CPU steal time is between 0 and 2. A 'WARNING' is displayed if it is 3 to 7; from 7 onwards, it will be in a 'CRITICAL' state. While conducting the experiments, there are some factors related to the display of these results; when the steal time percentage is between 0 and 2, it seems that everything runs smoothly without any delay; with a percentage increase between 3 and 7, the execution seems a bit delayed; so, it generates a WARNING display. Above seven percent, there is a high jitter in the execution; so, it is a CRITICAL situation. Listing 3 shows the CPU steal time case output, and a steal time status line is displayed.

Our proposed solution collects the monitoring data on the VM using the procfs, kernel pseudo-file system. All the virtual machines run on the compute node and work as the process of the cloud compute node. The information about the steal time of the VMs is mapped with the PID because each VM runs as a process in a cloud compute node. The information about the steal time is stored in the /proc folder. Our monitoring solution explores the /proc folders and fetches the steal time information. And to the best of our knowledge, the monitoring of the steal time of the VMs from the /**proc** folder has not yet been conducted.

**Listing 3**: Steal time status.

```
case $stealtime in
[0-2]*)
echo "OK—Steal time for VM $PID is $stealtime%."
exit 0
;;
[3-7]*)
echo "WARNING—Steal time for VM $PID is $stealtime%."
exit 1
;;
[7-*]*) echo "CRITICAL—Steal time for VM is $PID $stealtime%."
exit 2
;;
Esac
```

## 4. Implementation

We presented our proposed methodology for evaluating CPU steal time, encompassing various factors and utilizing virtual machines (VMs) within the experimental setup. Additionally, we provided a comprehensive overview of the data collection technique and the evaluation metrics employed in our study. To validate the effectiveness of our monitoring solution, we implemented an OpenStack-based cloud monitoring system specifically designed to monitor the CPU steal time of VMs. It was crucial to test the maximum capacity of our monitoring solution under real-world conditions.

To conduct our experiments, we established a cloud environment consisting of two Ubuntu servers, on which OpenStack was installed. One server was designated the cloud controller node, while the other was the compute node. Both nodes were configured identically and equipped with eight CPUs operating at a speed of 2.4 GHz, 8 GB of RAM, and 1 TB of ROM storage. Multiple VMs were launched within the cloud environment, surpassing the available physical resources. We utilized different operating system images for the VMs, including Ubuntu 20.04 LTS, CirrOS 16, and Fedora 38. Figure 2 provides a detailed list of the OS images used for booting the virtual machines. Through this experimental setup, we aimed to thoroughly examine and evaluate the CPU steal time of the VMs, considering various factors such as resource utilization and the impact of different operating system images. This allowed us to gather meaningful data and insights on VM performance and on the behavior of VMs with regard to CPU steal time.

```
m1@controller:~$ openstack image list
+--------------------------------------+----------------+--------+
| ID                                   | Name           | Status |
+--------------------------------------+----------------+--------+
| fa622a60-8c0d-4f49-83bc-cf61f36e7015 | fedora_gui     | active |
| 4c5b9667-d62a-41bb-9362-67e7fc56b280 | fedora         | active |
| e19e540f-a2fd-419d-a660-695a756cba84 | windows-server | active |
| 34bf1632-86ed-46ca-909e-c6ace830f91f | ubuntu-server  | active |
| bf6cdcdc-451a-463a-8927-4e09888e9537 | fedora-1       | active |
+--------------------------------------+----------------+--------+
```

**Figure 2.** List of images for VMs.

To implement our cloud monitoring solutions, we used the capabilities of OpenStack, an open source cloud computing platform. OpenStack is the foundation of our monitoring

infrastructure; it provides a scalable and flexible environment to accommodate varying workloads. In conjunction with OpenStack, we integrated two essential components: Nagios Core and NRPE (Nagios Remote Plugin Executor). Nagios Core is a powerful monitoring system that allows us to define and manage the monitoring of various resources and services within the cloud environment. NRPE plays a vital role in our monitoring setup, enabling the execution of Nagios plugins on remote hosts, allowing for real-time monitoring and data retrieval from different nodes in the cloud infrastructure. By combining OpenStack, Nagios Core, and NRPE, we created a seamless and robust monitoring solution that empowers the user to monitor critical performance metrics such as CPU usage, memory utilization, network traffic, and more. The implementation process is straightforward and resource-efficient, requiring minimal resources. The workflow of Nagios and NRPE can be seen in Figure 3.

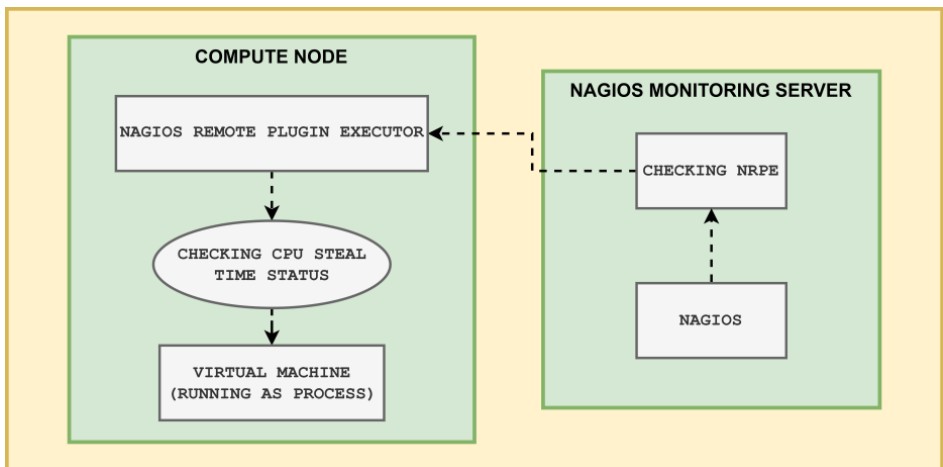

**Figure 3.** Nagios and NRPE workflow.

Figure 4 illustrates the running processes within our cloud monitoring environment. Each virtual machine (VM) is a separate process within the host operating system. To thoroughly evaluate the performance and behavior of the VMs, we subjected them to full load conditions. We created 10 VMs on our cloud compute node, with one core allocated to each VM. As it is has already mentioned that each compute node has eight cores, we overprovisioned our cloud resources. Under such circumstances, it is common for VMs to experience instances where they must wait for the physical CPU to become available. This wait time, or CPU steal time, is a crucial metric for assessing cloud performance. We executed a script in each VM to utilize a 100% CPU load. Examining Figure 4, we can observe that some VMs are not receiving 100% utilization of the CPU (for example, PID 8097 receives 57.8% of the CPU instead of 100% of the CPU. This discrepancy highlights the impact of CPU steal time on the VMs, indicating that resource contention may affect their performance. By monitoring and analyzing CPU steal time, we gain valuable insights into the efficiency and allocation of CPU resources among the VMs within our cloud monitoring system.

Each running process within our cloud compute nodes represents a specific virtual machine (VM), and the hypervisor undertakes the task scheduling among them. To gather information about these processes, we rely on the virtual file system, procfs, which provides insights into the system's internal state. Specifically, the steal time information related to the VMs can be obtained from the /proc/[pid]/folder. We developed a script, which was inserted into every VM within our cloud environment to facilitate the monitoring process. This script plays a crucial role in collecting the steal time information of each VM. By executing this script, we can accurately measure the steal time for each VM. The steal time data are presented as a percentage, indicating the portion of CPU time that has

been "stolen" or taken away from the VM. This allowed us to assess the impact of the CPU contention on the VMs' performance.

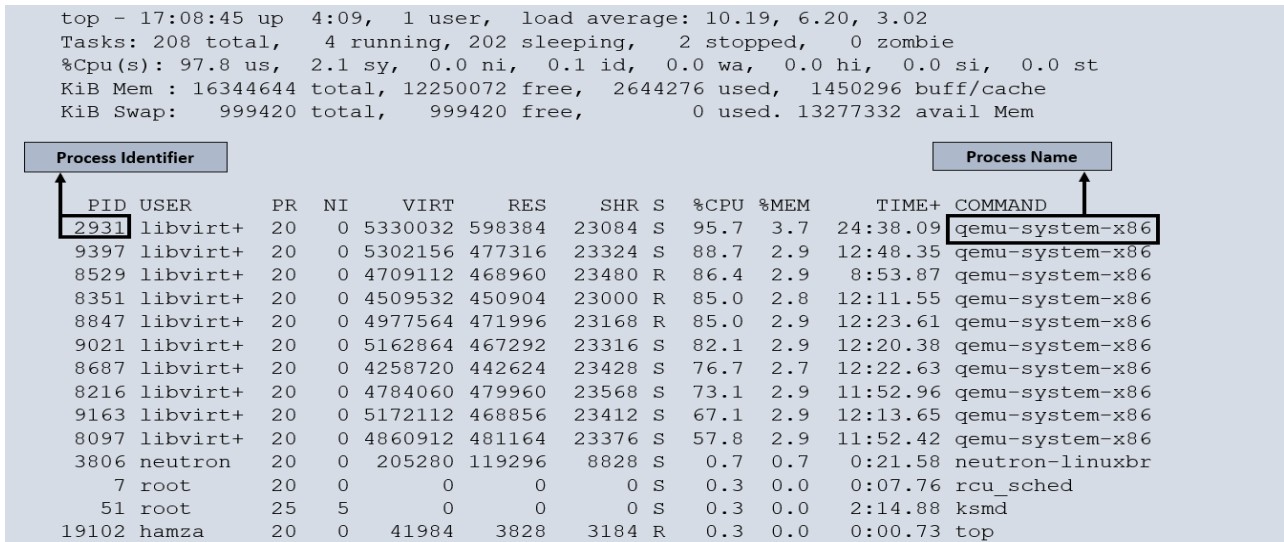

**Figure 4.** Running process on cloud computing node.

To ensure the seamless integration of our script into the VMs, we leveraged cloud-init, a tool that facilitates the automatic configuration of cloud instances during the VM boot process. By utilizing cloud-init, we can effortlessly insert our script into every VM, enabling efficient and consistent monitoring of steal time across the entire cloud environment.

The cloud compute node provides valuable information that can be identified using the process ID (PID) of the virtual machine (VM), as depicted in Figure 5. The creation of VMs involves executing instructions at the cloud controller node. Once the user initiates the VM creation process, the cloud compute node launches the VM. Subsequently, we performed a script to apply a load to each VM, ensuring that they operated at full capacity. This procedure was conducted for all the VMs, resulting in their running at full load.

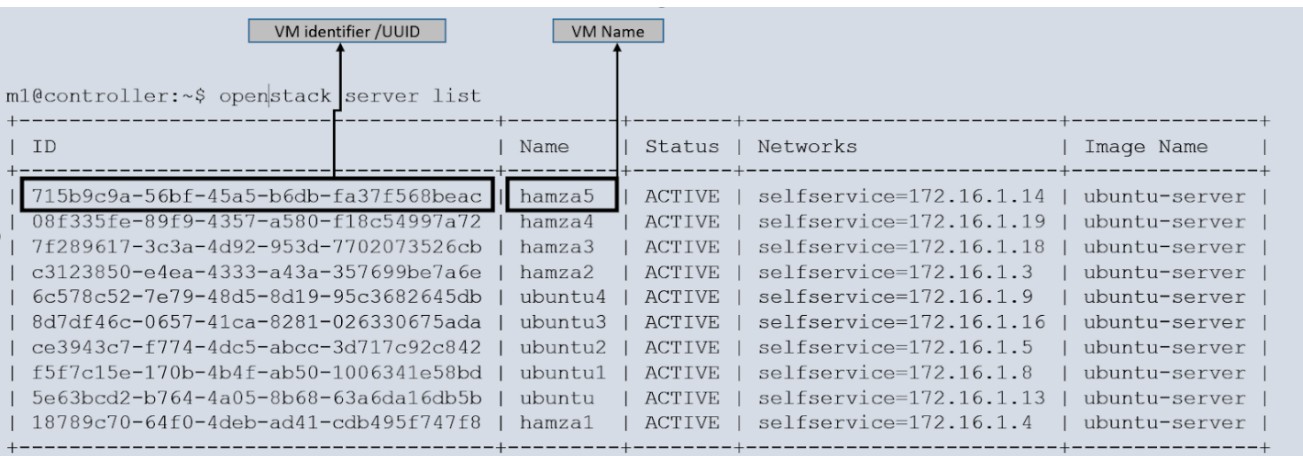

**Figure 5.** List of virtual machines on cloud controller node.

However, some VMs may not immediately receive the necessary CPU resources due to resource constraints, leading to waiting periods. This wait time is called steal time and can be accessed within the /proc/[pid] folder.

We gathered information regarding steal time from the VMs themselves. The code was responsible for monitoring the VMs executed across all instances, enabling us to obtain the CPU steal time of each VM as a percentage. This information was extracted from the

/proc folder within the VMs. The gathered data, which included stolen time information, were consolidated and stored in a file. An example of these monitored data is presented in Figure 6, where the VM status is indicated as "OK" when the steal time percentage remains below 2. The metric data reflect the steal time information in the rate, providing valuable insights into VM performance and resource allocations.

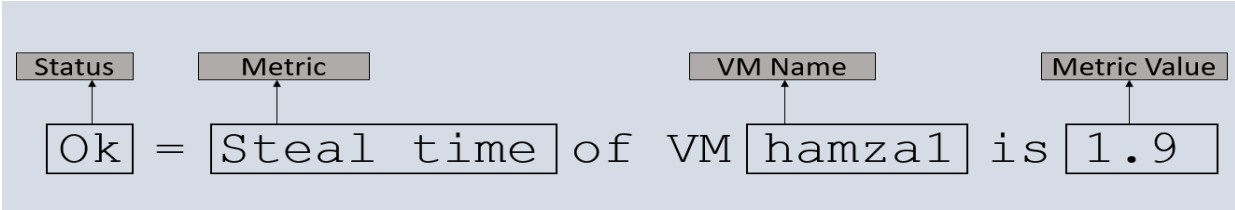

**Figure 6.** Dashboard VM status line for CPU steal time.

When all the virtual machines (VMs) within the cloud environment operate at full load, they inevitably experience CPU allocation delays. This occurs because, at any given time, all the VMs require access to the entire CPU resource. The hypervisor, which is responsible for scheduling and managing CPU allocation, assigns CPUs to the VMs. As a result, certain VMs have to wait for previous tasks to be completed by other VMs, leading to CPU contention. By utilizing the top command within the cloud compute node, we can observe the CPU utilization of the running VMs. This provides insights into the impact of running the VMs at full load, which inevitably leads to decreased CPU availability for each VM. The CPU percentage indicates the portion of CPU resources allocated to each VM. In contrast, the steal CPU represents the remaining CPU resources that are not accessible due to contention and wait times. The steal CPU percentage can be obtained by using Equation (1), i.e., CPU steal.

$$\text{Steal CPU} = 100 - \text{CPU percentage} \tag{1}$$

This steal CPU time is collected from all the VMs using a bash script executed on all the VMs and then combined with all the results of the CPU steal time. Figure 7 highlights the CPU steal times collected as percentages.

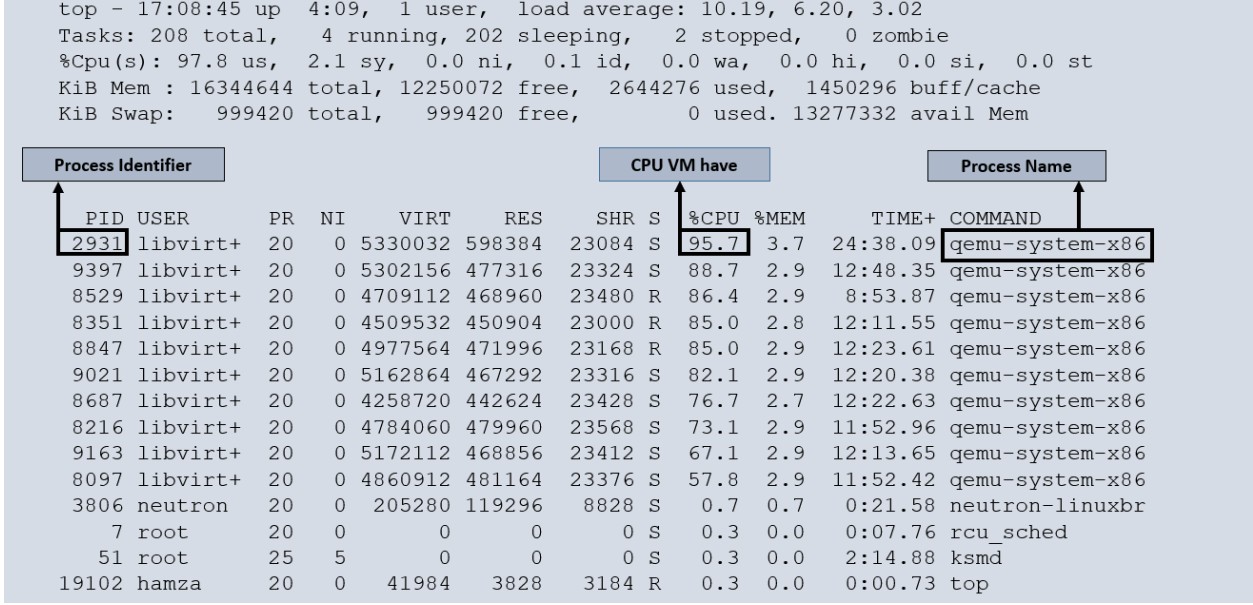

**Figure 7.** Results of CPU steal time.

Here, we highlight some metrics to evaluate our monitoring solution. These are the attributes addressed.

(i)    Monitoring latency

Linux's time utility plays a crucial role in measuring the latency of the executed script. To determine the latency, we initially ran the script on one of the virtual machines (VMs) operating at full load, while the remaining VMs had no limitations. The measured execution time for the script was 0.138 s. We repeated this experiment 30 times under the same setup, which resulted in an average execution time of 0.153 s for the code with no steal time. Next, we conducted another experiment where all 10 VMs ran with a total load. Despite overprovisioning with eight core CPUs, we launched 10 VMs and put them under full load. Ideally, each VM requires 100% of the CPU to run at full capacity. However, due to overprovisioning, some VMs fail to receive the entire CPU allocation, leading to wait times or steal time. These VMs must wait for the CPU to become available, resulting in increased execution delays. We ran the same script in this scenario, but the measured time now increased to 0.381 s. This extended execution time was a consequence of the CPU operating at full load, causing a delay in script execution. As with the previous experiment, we repeated this process 30 times and obtained an average execution time of 0.337 s. It is evident that with the entire load placed on the VMs, the execution time significantly increases, indicating the impact of steal time on the overall performance of the VMs.

(ii)    Monitoring Steal time of VM

To accurately assess the steal time factor of VMs, we leveraged the powerful Linux top command, which provides comprehensive metrics for VMs running on the cloud compute node. To streamline the process, we employed crontab to schedule the execution of our bash script, ensuring regular monitoring at two-minute intervals. The resulting data were then stored in a dedicated text file for further analysis.

To establish a baseline, we initially executed the script for an hour without imposing any load on the VMs. During this duration, we recorded an average steal time of 0.22 s. Subsequently, we proceeded to the main experiment by subjecting the VMs to a full load. By overprovisioning eight core CPUs and launching 10 VMs, we created a demanding environment to monitor steal time closely. The same script was executed for another hour under these conditions. As anticipated, the VM steal time experienced a substantial increase, with an observed average steal time of 5.9 s. We relied on OpenStack, a robust cloud management platform, within a real cloud environment for the actual implementation. By launching 10 VMs, each operating on an eight-core CPU, we simulated a comprehensive workload to monitor steal time effectively. In terms of evaluation, we collected data on monitoring latency and CPU steal time. These evaluations provide valuable insights into the delays that arise when VMs operate under a full load, shedding light on the impact of resource allocation on steal time.

## 5. Results and Discussion

Our primary objective revolves around monitoring latency and understanding the impact of full load on steal time when all the VMs are running. To achieve this, we conducted a series of experiments to obtain the desired monitoring results. Based on our extensive evaluations, we present the empirical assessment of our monitoring solution. Furthermore, we delve into the evaluation results regarding the performance of VMs under different load conditions. We evaluated the performance of our monitoring solution by launching several VMs on the cloud compute node and monitoring each VM with varying loads. To ensure a comprehensive utilization of resources, we overprovisioned the cloud compute node, effectively leveraging all available cores with a full load to measure the steal time factor. By representing the data graphically, we can effectively assess the performance of our monitoring solution.

To collect monitoring data, we relied on Linux's top utility. This powerful program captures a plethora of instructions related to VMs. To validate the accuracy of our monitoring

solution, we explored the top program, extracted the relevant information, and presented it in the form of graphical representations. This information includes the execution time of the steal time script on VMs with and without a load. In Figure 8, we showcase a graph that compares and contrasts the experiments conducted on the VMs. The red line represents the execution time of the steal time script with all the VMs running on full load, while the blue line signifies the scenario where no limitations are imposed on the VMs. The *y*-axis indicates the time in seconds for script execution, while the *x*-axis denotes the intervals at which the data were collected.

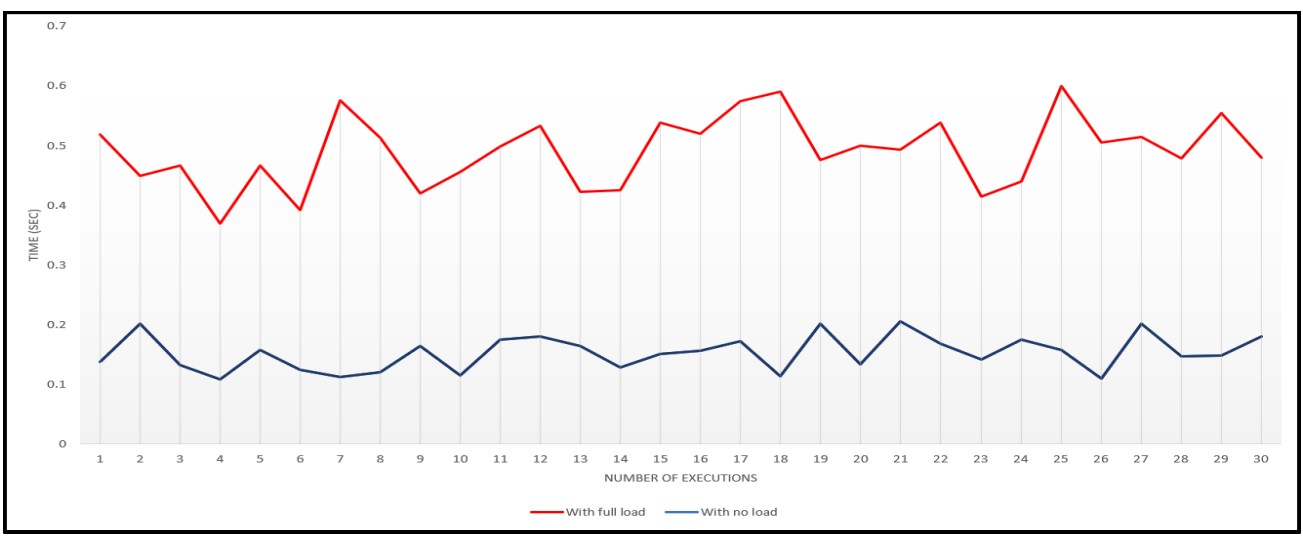

**Figure 8.** Monitoring latency.

Upon analyzing the graph, we observe that the blue line exhibits minimal variation, representing the scenario with no load on the VMs. This indicates that the execution time of the script is relatively consistent. Throughout the intervals, the execution time generally falls within the range of 0.1 to 0.2 s.

Conversely, when the entire load is applied to the VMs, it significantly impacts the execution time of the code. Depending on the availability of the VM's physical CPU, the execution time fluctuates between 0.4 and 0.6 s. This demonstrates that the execution time of the script is directly influenced by the load imposed on the VM.

We conducted monitoring of the CPU steal time on all ten virtual machines using our script. The evaluation results are presented in Figure 9. The *x*-axis represents the steal time percentage, while the *y*-axis indicates the number of iterations. With the VMs running on full load, we observe a prominent trend in the red line, indicating the steal time percentage. A higher value on the red line indicates that a significant portion of the CPU is unavailable during that interval, resulting in increased steal time. From Equation (2) (available CPU in %), we can observe that.

$$\text{Available CPU (\%)} = 100 - \text{Steal time (\%)}. \tag{2}$$

Figure 9 provides valuable insights into the relationship between CPU availability and steal time for VMs. As a VM lacks an entire CPU, its steal time increases. However, the steal time gradually reduces when the hypervisor schedules tasks and allocates available CPUs to the VM. This pattern holds for all the VMs in the system. On the other hand, the blue line in the graph represents a scenario with little to no steal time, indicating that there is no load on the VMs and that each VM receives its desired CPU allocation.

Our monitoring solution's results were thoroughly discussed and presented through graphical representations. The execution delay of the script is directly influenced by the load imposed on the VMs. When running with a full load, the script requires more time for execution compared to the scenarios with no limitations on the VMs. Furthermore, a higher

steal time percentage on a VM leads to increased delay in script execution. The script is executed across all the VMs, and the results are calculated individually for each VM. By analyzing the behavior of VMs under full load, we can conclude that the steal time of VMs increases until the CPU becomes available for each VM. This pattern repeats for all VMs at different instances in time. These findings indicate that the steal time factor intensifies with increased load. Moreover, our monitoring solution proved to be efficient in accurately monitoring VM steal time without imposing any additional overheads on the system.

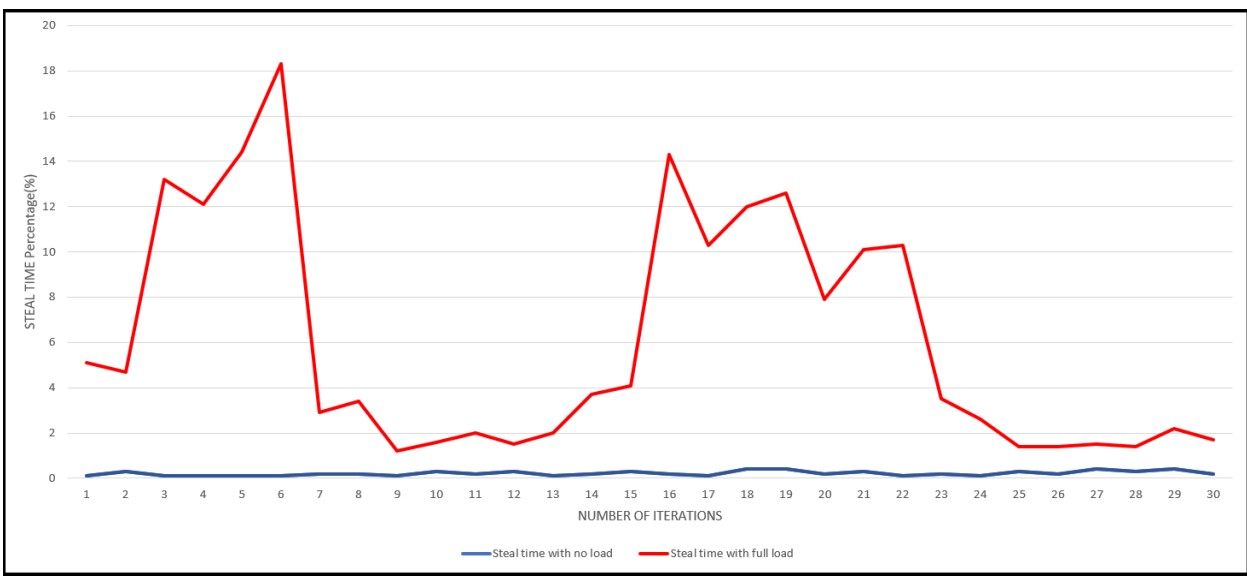

**Figure 9.** Steal time percentage.

## 6. Conclusions

In this research, our primary objective was to monitor cloud performance through the execution of virtual machines (VMs) under full load conditions. We successfully achieved our aims and objectives and demonstrated the significance of this research in cloud monitoring. To accomplish our goals, we extensively reviewed state-of-the-art cloud monitoring solutions. Based on this review and related work, we proposed a novel solution for monitoring CPU steal time in an OpenStack-based cloud monitoring environment. We emphasized the importance of steal time monitoring as a critical factor in evaluating cloud performance. We implemented our framework in an actual cloud environment to evaluate the effectiveness of our proposed OpenStack-based cloud monitoring system. We established an OpenStack-based cloud with a controller node and a compute node. To measure the performance of our monitoring solution, we introduced a script into the VMs to monitor their CPU steal time. During our experimentation, we observed the monitoring latency before and after applying a full load to the VMs. We also measured the steal time of the VMs. The results indicated that the CPU steal time of a VM increases until it receives the CPU it is waiting for. Moreover, we observed a significant impact on the steal time when running the VMs under a full load compared to running them without limitations. In the absence of a burden, the steal time was found to be minimal.

Our proposed solution for monitoring the CPU steal time of VMs proved to be highly effective and efficient. We believe this aspect of VM monitoring has been overlooked in previous research studies. The method employed to collect CPU steal time data from VMs is robust and does not impose an excessive burden on the VMs. In summary, our monitoring solution offers a lightweight, efficient, and scalable approach to cloud monitoring. It provides valuable insights into the performance of VMs, particularly with regard to CPU steal time. We implemented it in the private cloud, and in the future, we can implement it in the public and hybrid clouds. We filled a research gap by introducing and successfully implementing this novel monitoring approach.

**Author Contributions:** Conceptualization, H.A. and H.J.S.; methodology, H.A., H.J.S. and A.S.; validation, A.O.I., M.A. and M.E.; investigation, A.O.I., M.A. and M.E.; resources, H.A., H.J.S. and A.S.; writing—original draft preparation, H.A., H.J.S. and A.S.; writing—review and editing, all authors; supervision, H.J.S.; project administration, H.J.S., A.O.I., M.A. and M.E.; funding acquisition, M.A. and M.E. All authors have read and agreed to the published version of the manuscript.

**Funding:** This project is funded by Princess Nourah bint Abdulrahman University Researchers Supporting Project number (PNURSP2023R383), Princess Nourah bint Abdulrahman University, Riyadh, Saudi Arabia.

**Institutional Review Board Statement:** Not applicable.

**Informed Consent Statement:** Not applicable.

**Data Availability Statement:** Not applicable.

**Acknowledgments:** This work was supported through Princess Nourah bint Abdulrahman University Researchers Supporting Project number (PNURSP2023R383), Princess Nourah bint Abdulrahman University, Riyadh, Saudi Arabia.

**Conflicts of Interest:** The authors declare that they have no conflict of interest.

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
