# Peer review of "Exploring Performance Degradation in Virtual Machines Sharing a Cloud Server"

_applsci, doi:10.3390/app13169224_

Round 1

Reviewer 1 Report

The authors delve into the crucial aspect of CPU steal time, which quantifies the duration that a virtual CPU spends awaiting processing by the physical CPU.  The current paper presents deficiencies to improve, such as:

i) The Conclusions and Abstract must be self-contained.

ii) Improve related work

iii) The solution should be described more clearly for the reader. Similarly in the implementation section and the algorithms presented.

iv) Improve the analysis of the results obtained and compare them with other methods.

v) Highlight the novelty and importance of addressing this problem, and the proposed solution.

Best Regard

For the moment, Moderate editing of English language required.

Author Response

Response To Reviewer #1

We want to thank you for your positive feedback. Your detailed comments have considerably helped improve the revised manuscript's clarity.

Reviewer Comment # 1

Response

Thank you for your valuable comment. In the updated manuscript, we have updated our abstract and conclusion.

Reviewer Comment # 2

Response

We are grateful for the respectable reviewer on encouraging comments and thankful for their insightful comments that helped us improve our paper's quality. We have improved our related work by adding more research articles which can be seen in Section 2, i.e., Related Work.

Reviewer Comment # 3

Response

 Thank you for your valuable comment. In the updated manuscript, we have improved the implementation and algorithm presented in the manuscript. We try our level best to add more clarity to the readers.

Reviewer Comment # 4

Response

 Thank you very much for thoroughly reviewing our paper and constructive comments. To our best knowledge, none of the other work has monitor CPU steal time, so it is impossible to compare with other works.

Reviewer Comment # 5

Response

We appreciate you bringing this up. We concur with this statement. We have mentioned in the introduction that CPU steal is one of the essential factors in cloud monitoring. To the best of our knowledge, none of the others monitored CPU steal time for cloud service. In this work, we have monitored and proved that CPU steal time is one of the most critical monitoring factors in the cloud.

Reviewer 2 Report

This manuscript examines the repercussions of CPU stealing time on the performance of virtual machines.

I’d like to suggest authors change the title to better fit with the paper's content.

For the statement below, could add to the text some examples and provide the references as well.

Cloud monitoring systems use agents, also called intrusive cloud monitoring solutions (e.g., x,yz)?

As monitoring evolved a lot in the past few years, I’d like to see in the related work section some recent studies from 21, 22, or even 23. Still, the mention of centralized and decentralized cloud monitoring is really interesting and matches with new trends like blockchain technology. Based on that, the author could bring some more details to this topic. Finally, based on the above comments, I’d like to see a comparative table with the name of the tools, features, and characteristics related to them, as well as a discussion that drives the true need for stealing time monitoring. I think it is imperative for the authors to highlight the manuscript from other studies since it is a well-studied topic in the field.

The proposal section is not well designed and needs to be improved.

It is not clear how the authors integrated the stealing time monitoring on top of OpenStack. Please, describe the OpenStack monitoring module and also illustrate the architecture model and how it integrates with OpenStack, and how the flow operates on that.

Section 4 and 5 were not well designed and needs to be improved. 

Ubuntu 16 is outdated, why not use a new OS?

Which are the tools and loads used in the experiments? Stress-ng or another? Please, create a table and describe it in detail.

Could create a chart crossing stealing time for each VM during the experiment period? I think It could be interesting to observe VM behavior under stressful conditions. 

Could the authors replicate this strategy with some Open Stack nodes in order to create a realistic test scenario? 

Usually, we did not use the full server for placing VMs. It means latency in Figure 7 could steady near zero if you reserve at least 10% of the server resources for system administration. 

in the conclusion, the authors must indicate which are the real use cases of the proposed monitoring. How does this kind of monitoring help in MV placement and scheduling? Still, is it possible to integrate this tool with other monitoring systems? Which are the next steps?

Overall comments.

Please, verify all the calls for the figures and replace them, for instance:

From figure 1 to Figure 1, always use uppercase for this kind of call or reference, same for Algs, Listings, and so on.

Please, apply the same spacing and size for Alg 1 and 2

Remove bold from listings

Convert Listing 3.3 for math design 

Convert Figures 2, 3, and 4 to tables.

Convert Figure 5 to a listing

Convert Lines 364, and 441 to a formula

In the following statement, Figure 8 below, remove the word below.

Author Response

Response To Reviewer #2

We want to thank you for your positive feedback. Your detailed comments have considerably helped improve the revised manuscript's clarity.

Reviewer Comment # 1

Response

Thank you for your valuable comment. In the updated manuscript, we have updated the Title of the Paper.

Reviewer Comment # 2

Response

We are grateful for the respectable reviewer on encouraging comments and thankful for their insightful comments that helped us improve our paper's quality. We have updated the line which was commented on with suitable citations.

Reviewer Comment # 3

Response

 We appreciate you bringing this up. We concur with this statement. In the updated manuscript Section 2, i.e., Related Word, we have added some used research articles from recent years. Also, we have added the comparative table, which contains the tools, their features, and related characteristics.

Reviewer Comment # 4

Response

 Thank you very much for thoroughly reviewing our paper and constructive comments. In the updated manuscript, we have updated the proposed section.

Reviewer Comment # 5

Response

Thank you for your valuable comment. On lines 421-425, we discussed leveraging OpenStack by combining it with the Nagios Core and NRPE (Nagios Remote Plugin Executor).

Reviewer Comment # 6

Response

 Thank you very much for thoroughly reviewing our paper and constructive comments. We have updated Sections 4 and 5 in the manuscript.

Reviewer Comment # 7

Response

 In the updated manuscript, We have done it on the new Ubuntu version, i.e., Ubuntu 20. CPU steal time is a feature of cloud service providers, so it is not dependent on the Ubuntu version.

Reviewer Comment # 7

Response

We appreciate you bringing this up. We concur with this statement. We used the following script to generate 100% CPU load on each VM

# yes > /dev/null &

Reviewer Comment # 8

Response

 Thank you for your valuable comment. CPU switches context so fast that it cannot be captured. Our primary focus is to capture the CPU steal time and highlight its importance.

Reviewer Comment # 9

Response

We appreciate you bringing this up. We have created a real OpenStack-based cloud, and all the implementation is done.)

Reviewer Comment # 10

Response

 Thank you for pointing this out. We agree with this comment. We hypothesize that cloud service providers run CPU on full load. For example, if a server has ten physical cores, cloud providers usually provision 20 VM (assuming one core for each) on that server. In Figure 7, we have highlighted that our proposed solution is lightweight and does not put any noticeable latency on monitoring.

Reviewer Comment # 11

Response

Thank you for the valuable comment. It is very crucial to measure CPU steal time; if the cloud service provider notices that due to some VM machines (running on full CPU load), the performance of other VM machines on the same host is degraded, they can migrate those VMs (on full CPU load) to other hosts. Yes, it is possible, and we have implemented our script using NAGIOS. Cloud Service Providers can use our solution to measure the CPU steal time.

Reviewer Comment # 12

Response

 Thank you very much for thoroughly reviewing our paper and constructive comments. We have updated this comment in the manuscript.

Reviewer Comment # 13

Response

We have added the same spacing in the Algorithms and removed all the bold from the listings. Thank you for pointing this out.

Reviewer Comment # 14

Response

We appreciate you bringing this up. It is a bash script. We do not understand what is confusing in it. Kindly further elaborate on it.

Reviewer Comment # 15

Response

These are the print screens of the server, so the figure is more suitable in this regard.

Reviewer Comment # 16

Response

We appreciate you bringing this up. We concur with this statement. It is the output of our dashboard. That's why we have marked it as a figure.

Reviewer Comment # 17

Response

We have updated a comment in the manuscript.

Reviewer 3 Report

I have finished my review of the submitted manuscript and below are my comments:

1) "By offering valuable insights, our analysis sheds light on the significance of monitoring CPU steal time and its profound impact on cloud performance"

Describe the insights insinuated above.

2) Contributions 2 and 3 need to be described in detail to bring out the particulars of the proposed solution and how it compares with other existing solutions. This way, the contributions of this work will be clear to the readers.

3) The quality of English needs to be improved, e.g., "Several issues are there in monitoring cloud network resources.." 

There are so many instances of such mistakes.

4) The essence of Section 2, "Related work" is to establish what has been done and try to bring out the gaps on what has been done so far. This critique of the current work is largely missing in this section.

5) In line 131, the authors write as follows: "The study says that there is no way this loss can be monitored (i.e., CPU steal time)"

Which study is being implied above? In addition, this should be framed as follows:  " The study by [] shows/demonstrates that..."

6) The clarity of all figures need to be improved.

7) The authors need to introduce the mathematical formulations for the latency and Steal time Percentage.

8) More comparative evaluations need to be included in Section 5

9) The Conclusion section needs to be written in continuous prose. In addition, future research work in this domain must be included in this section.

10) In line 479 and 480, the authors write as follows: "Our monitoring solution is a lightweight, efficient, and scalable cloud monitoring solution."

Where are the results for lightweight, efficiency and scalability implied above?. Ensure that all these results are included in Section5. Thereafter, validate the above statement with empirical results you obtained from the experimentations carried out.

11) In the list of references, articles published in the year 2021, 2022 and 2023 are missing. Ensure you update your work with these current works.

The quality of English used throughout the paper is poor and needs extensive editing.

Author Response

Response To Reviewer #3

We want to thank you for your positive feedback. Your detailed comments have considerably helped improve the revised manuscript's clarity.

Reviewer Comment # 1

Response

Thank you for your valuable comment. In the Introduction Section, i.e., Section 1, On lines 67-80, we have already discussed the importance of CPU steal time and its impact on the cloud performance while doing tasks.

Reviewer Comment # 2

Response

We are grateful for the respectable reviewer on encouraging comments and thankful for their insightful comments that helped us improve our paper's quality. We have added a comparative table in Section 2 so that the contributions of this work will provide more clarity to the readers.

Reviewer Comment # 3

Response

 Thank you for your valuable comment. In the updated manuscript, we improved the quality of English.

Reviewer Comment # 4

Response

 Thank you very much for thoroughly reviewing our paper and constructive comments. We have tried to highlight CPU steal time and added the comparative table in section 2, i.e., Related work.

Reviewer Comment # 5

Response

We appreciate you bringing this up. We concur with this statement. On line 167, We have cited the paper. It is not our claim; it is mentioned in the study.

Reviewer Comment # 6

Response

We appreciate you bringing this up. We make sure that all the figures are of high quality.

Reviewer Comment # 7

Response

We appreciate you bringing this up. We concur with this statement. We have mentioned in the introduction that CPU steal is one of the essential factors in cloud monitoring. We have already provided our algorithms. Mathematical modeling is out of the scope of this work.

Reviewer Comment # 8

Response

We appreciate you bringing this up. As we already mentioned, none of the existing works monitored the CPU steal time. So we cannot compare the results with other works.

Reviewer Comment # 9

Response

Thank you for pointing this out. We agree with this comment. We have updated our conclusion along with future work in this domain, as seen in section 6.

Reviewer Comment # 10

Response

We are grateful for the respectable reviewer on encouraging comments and thankful for their insightful comments that helped us improve our paper's quality. Our monitoring solution is a lightweight solution that can be visualized from Figure 7(Monitoring Latency) that on full load, the execution time fluctuates between 0.4 to 0.6 seconds. The efficiency of our solution can be seen in Figure 8 (Steal time), in which we have figured it out using full load and without a full load. The solution's scalability depends upon the communication models, i.e., Push and Pull, discussed on lines 252-262.

Reviewer Comment # 11

Response

Thank you for pointing this out. We agree with this comment. We have added some research articles in section 2, i.e., Related Work from the said years.

Round 2

Reviewer 1 Report

The authors delve into the aspect of CPU steal time, quantifying the time a virtual CPU spends awaiting processing by the physical CPU. There are improvements in the current version, but you should dig deeper into the following points.

i) Include current situation diagram vs. proposed solution diagram

ii) Include a diagram or process flow of how to implement the proposed solution.

iii) Quantify by indicators of the performance obtained of the proposal. It should be included to compare the current and proposed situations. That is, % of improvement, overload, and time, among others.

Regards. The reviewer

Moderate editing of English language required

Author Response

Message from the Authors

Dear Editors and Reviewers,

We appreciate your helpful feedback, which helped us raise the quality of the manuscript. In the updated paper, we have answered your comments and taken into account your insightful ideas, emphasizing, in particular, the major contributions of this work.

In the following thorough answer, we address each comment thoroughly. Following each of the boxed remarks that we received, we have written our reply. We trust you will find the amended manuscript to your satisfaction and look forward to hearing from you.

Sincerely Yours

Response To Reviewer # 1

We want to thank you for your positive feedback. Your detailed comments have considerably helped improve the revised manuscript's clarity.

Reviewer Comment # 1

Include current situation diagram vs. proposed solution diagram

Response

Thank you for your valuable comment. The scope of this work is to demonstrate the significance and importance of CPU steal time, which is prominent and essential. We have thoroughly examined the prolonged waiting period of the CPU caused by steal time. However, it is important to note that we have not explicitly addressed its impact on Process degradation or Application degradation as it falls outside the scope of this study. In general, our implementation encompasses the entirety of the CPU steal time analysis.

Reviewer Comment # 2

 Include a diagram or process flow of how to implement the proposed solution

 Response

We are grateful for the respectable reviewer on encouraging comments and thankful for their insightful comments that helped us improve our paper's quality.  In section 3, we have already included the Schematic diagram of the proposed solution along with the process flow as to how to implement the proposed solution, which can be seen on lines 266-279. Also, in section 4, we discussed the implementation flow of OpenStack along with Nagios, which is available on lines 421-433.

 Reviewer Comment # 3

 Quantify by indicators of the performance obtained of the proposal. It should be included to compare the current and proposed situations. That is, % of improvement, overload, and time, among others.

Response

Thank you for your valuable comment. This research aims to illustrate CPU steal time's value and significance, which is crucial and noticeable. We have carefully examined the prolonged CPU wait time brought on by steal time. It is essential to highlight that because it is outside the purview of this study, we have not explicitly addressed its influence on any specific process and application degradation, which leads to a comparison of the % of improvement, overload, and time among the current and proposed situation. Our solution, in general, covers the complete CPU steal time analysis.

Reviewer 2 Report

Please,

Insert your work in the comparative table!

The equations are not using the correct schema. Please review Journal examples.

Fig 5 should be converted to an equation.

Please, remove spacing from all algorithms.

Please, design proper tables instead of Figures for Figures 2, 3, and 4. Still, it only presents useful information.

Author Response

Message from the Authors

 Dear Editors and Reviewers,

We appreciate your helpful feedback, which helped us raise the quality of the manuscript. In the updated paper, we have answered your comments and taken into account your insightful ideas, emphasizing, in particular, the major contributions of this work.

In the following thorough answer, we address each comment thoroughly. Following each of the boxed remarks that we received, we have written our reply. We trust you will find the amended manuscript to your satisfaction and look forward to hearing from you.

Sincerely Yours

Response To Reviewer #2

We want to thank you for your positive feedback. Your detailed comments have considerably helped improve the revised manuscript's clarity.

Reviewer Comment # 1

Insert your work in the comparative table!

Response

Thank you for your valuable comment. We have added our work in the comparative table in the updated manuscript.

Reviewer Comment # 2

The equations are not using the correct schema. Please review Journal examples.

Response

We are grateful for the respectable reviewer on encouraging comments and thankful for their insightful comments that helped us improve our paper's quality. We have changed the equations to the correct schema in the updated manuscript.

Reviewer Comment # 3

Fig 5 should be converted to an equation.

Response

 We appreciate you bringing this up. We concur with this statement. It is the output of our monitoring dashboard, it is not an equation. That's why we have marked it as a figure.

Reviewer Comment # 4

Please, remove spacing from all algorithms.

Response

 Thank you very much for thoroughly reviewing our paper and constructive comments. In the updated manuscript, we have removed the spacing from all the algorithms.

Reviewer Comment # 5

Please, design proper tables instead of Figures for Figures 2, 3, and 4. Still, it only presents useful information.

Response

Thank you for your valuable comment. These figures are the print screens of the server, so the figure is more suitable in this regard. Also, these figures show essential information regarding OS, VMs, and more, so we cannot edit it.

Reviewer 3 Report

I have gone through the revised manuscript and it is clear that the authors ignored to address most of my previous comments. In particular, the following issues are yet to be addressed:

1)In the abstract, the authors write as follows: "In this study, we focus on the impact of CPU steal time on virtual machine performance and the potential problems that can arise."

The authors need to state the results obtained for the impact of CPU steal time on virtual machine performance. They also need to illustrate some of the potential problems that can arise as a result of CPU steal time.

2) "Our analysis provides insights into the importance of CPU steal time monitoring and its impact on cloud performance."

The authors need to briefly describe the insights insinuated above. They also need to elaborate the insinuated impact of CPU steal time monitoring on cloud performance.

3)Contributions 2 and 3 need to be described in detail to bring out the particulars of the proposed solution and how it compares with other existing solutions. This way, the contributions of this work will be clear to the readers. In addition, the results for the comparative analysis alluded to in Contribution 3 need to be stated.

4)The quality of English needs to be improved, e.g., "Several issues are there in monitoring cloud network resources.." 

There are so many instances of such mistakes.

5)The clarity of all figures need to be improved.

6)The authors need to introduce the mathematical formulations for the latency and Steal time Percentage.

7)More comparative evaluations need to be included in Section 5

8) In the Conclusion section, the authors claim as follows: "..Our proposed solution for monitoring the CPU steal time of VMs has proven to be highly effective and efficient."

The above claim need to be supported by empirical results for effectiveness and efficiency.

9) In the Conclusion section, the authors write as follows: "Our monitoring solution is a lightweight, efficient, and scalable cloud monitoring solution."

Where are the results for lightweight, efficiency and scalability implied above?. Ensure that all these results are included in Section5. Thereafter, validate the above statement with empirical results you obtained from the experimentations carried out.

10) Some of the references used in this paper are too old. You should incorporate recent and relevant works.

Extensive editing of English language is required

Author Response

Message from the Authors

Dear Editors and Reviewers,

We appreciate your helpful feedback, which helped us raise the quality of the manuscript. In the updated paper, we have answered your comments and taken into account your insightful ideas, emphasizing, in particular, the major contributions of this work.

In the following thorough answer, we address each comment thoroughly. Following each of the boxed remarks that we received, we have written our reply. We trust you will find the amended manuscript to your satisfaction and look forward to hearing from you.

Sincerely Yours

Response To Reviewer #3

We extend our heartfelt gratitude for your positive feedback. Your in-depth comments have been immensely valuable in enhancing the clarity of the revised manuscript. We deeply appreciate the time and effort you invested in providing such valuable insights. However, we would like to apologize for any inconvenience caused by a formatting issue that resulted in some of the comment responses not being visible to you. Rest assured, we are actively addressing this matter to ensure all comment responses are made fully accessible. Once again, thank you for your valuable contribution, which has been pivotal in refining our work. Your support and feedback are highly valued as we strive to deliver a more comprehensive and polished manuscript.

Reviewer Comment # 1

In the abstract, the authors write as follows: "In this study, we focus on the impact of CPU steal time on virtual machine performance and the potential problems that can arise."

The authors need to state the results obtained for the impact of CPU steal time on virtual machine performance. They also need to illustrate some of the potential problems that can arise as a result of CPU steal time.

Response

Thank you for your valuable comment. We appreciate you bringing this up. In section 4, figure 4, it is clearly observed that we have created 10 VMs, each with one core. The compute node consists of 8 cores, leading to overprovision in our cloud resources. Our analysis shows that not a single VM is 100% using CPU utilization due to the CPU steal time. In general, this CPU steal time issues cause CPU degradation, which also causes different performance degradation of processes and application that will be running on it.

Reviewer Comment # 2

"Our analysis provides insights into the importance of CPU steal time monitoring and its impact on cloud performance."

The authors need to briefly describe the insights insinuated above. They also need to elaborate the insinuated impact of CPU steal time monitoring on cloud performance.

Response

We sincerely appreciate your valuable comment and thank you for bringing this matter to our attention. Figure 4 shows that we have deployed 10 VMs, each configured with a single core. However, the compute node within our cloud infrastructure possesses 8 cores, resulting in overprovisioning of resources. Our analysis has shown that due to CPU steal time, none of the VMs are utilizing the CPU to its full capacity, even though they are not reaching 100% CPU utilization. This issue of CPU steal time leads to CPU degradation, which, in turn, impacts the performance of various processes and applications running on the VMs.

Reviewer Comment # 3

Contributions 2 and 3 need to be described in detail to bring out the particulars of the proposed solution and how it compares with other existing solutions. This way, the contributions of this work will be clear to the readers. In addition, the results for the comparative analysis alluded to in Contribution 3 need to be stated.

Response

We express our gratitude to the esteemed reviewer for their encouraging comments and insightful feedback, which significantly contributed to enhancing the quality of our paper. Regarding contribution 2, we have addressed this aspect in Section 4, specifically in lines 438-449, where we elaborate on Figure 4. This figure illustrates the methodology used to measure CPU steal time, clearly demonstrating that under full load conditions, our VMs do not reach full CPU utilization capacity. As for contribution 3, we have introduced a comparative table in Section 2, facilitating a direct comparison between our proposed solution and existing ones. This addition enables readers to gain a better understanding of the unique advantages and distinctions of our work, promoting greater clarity and context for the audience.

Reviewer Comment # 4

The quality of English needs to be improved, e.g., "Several issues are there in monitoring cloud network resources.."

Response

 Thank you for your valuable comment. We have improved the quality of English in the updated manuscript.

Reviewer Comment # 5

The clarity of all figures need to be improved

Response

We appreciate you bringing this up. Most of the figures are the print screens of the server, we have tried our level best to attached high quality images.

Reviewer Comment # 6

The authors need to introduce the mathematical formulations for the latency and Steal time Percentage.

Response

We appreciate you bringing this up. We concur with this statement. We have mentioned in the introduction that CPU steal is one of the essential factors in cloud monitoring. We have already provided our algorithm 1 and algorithm 2. Mathematical modeling is out of the scope of this work.

Reviewer Comment # 7

More comparative evaluations need to be included in Section 5

Response

Thank you for raising this point, and we genuinely appreciate your input. As we previously stated, the monitoring of CPU steal time has not been addressed in any existing works. Therefore, direct comparisons of our results with those of other works in this specific context are not possible.

Reviewer Comment # 8

In the Conclusion section, the authors claim as follows: "..Our proposed solution for monitoring the CPU steal time of VMs has proven to be highly effective and efficient."

The above claim need to be supported by empirical results for effectiveness and efficiency.

Response

We greatly appreciate your observation, and we fully concur with your comment. In Section 5, the efficiency and effectiveness of our solution are demonstrated through Figure 9, where we present the analysis of Steal time under both full load and no full load conditions. Through our monitoring script, we have diligently tracked CPU steal time on all ten virtual machines, despite the overprovisioning of cloud resources, considering the compute node's 8 cores. As depicted in Figure 9, when the virtual machines are subjected to a full load, CPU steal time occurs, resulting in the VMs not reaching full 100% CPU utilization. The red line distinctly illustrates the percentage of steal time. Similarly, Figure 7 reinforces this observation, indicating that the VMs do not achieve 100% CPU utilization due to Steal time.

Reviewer Comment # 9

In the Conclusion section, the authors write as follows: "Our monitoring solution is a lightweight, efficient, and scalable cloud monitoring solution."

Where are the results for lightweight, efficiency and scalability implied above?. Ensure that all these results are included in Section5. Thereafter, validate the above statement with empirical results you obtained from the experimentations carried out.

Response

We are grateful for the respectable reviewer on encouraging comments and thankful for their insightful comments that helped us improve our paper's quality. Our monitoring solution is a lightweight solution that can be visualized from Figure 8(Monitoring Latency) that on full load, the execution time fluctuates between 0.4 to 0.6 seconds. The efficiency of our solution can be seen in Figure 9 (Steal time), in which we have figured it out using full load and without a full load. The solution's scalability depends upon the communication models, i.e., Push and Pull, discussed on lines 252-263.

Reviewer Comment # 10

Some of the references used in this paper are too old. You should incorporate recent and relevant works.

Response

We acknowledge and agree with this comment. While it is true that some of the cited references may be dated, they hold significant importance for our work and provide valuable support to our research. Therefore, including these citations is crucial to establish a solid foundation and strengthen the credibility of our study.
